# Revisiting Explicit Regularization in Neural Networks for Reliable Predictive Probability

## Abstract

From the statistical learning perspective, complexity control via explicit regularization is a necessity for improving the generalization of overparameterized models, which deters the memorization of intricate patterns existing only in the training data. However, the impressive generalization performance of overparameterized neural networks with only implicit regularization challenges the importance of explicit regularization. Furthermore, explicit regularization does not prevent neural networks from memorizing unnatural patterns, such as random labels. In this work, we revisit the role and importance of explicit regularization methods for generalization of the *predictive probability*, not just the generalization of the 0-1 loss. Specifically, we analyze the possible cause of the poor predictive probability and identify that regularization of predictive confidence is required during training. We then empirically show that explicit regularization significantly improves the reliability of the predictive probability, which enables better predictive uncertainty representation and prevents the overconfidence problem. Our findings present a new direction to improve the predictive probability quality of deterministic neural networks, which can be an efficient and scalable alternative to Bayesian neural networks and ensemble methods.

## 1 Introduction

As deep learning models have become pervasive in real-world decision-systems, the importance of producing a reliable predictive probability is increasing. In this paper, we call predictive probability reliable if it is well-calibrated and precisely represents uncertainty about its predictions. The calibrated behavior refers to the ability to match its predictive probability of an event to the long-term frequency of the event occurrence (Dawid, 1982). The reliable predictive probability benefits many downstream tasks such as anomaly detection (Malinin & Gales, 2019), classification with rejection (Lakshminarayanan et al., 2017), and exploration in reinforcement learning (Gal & Ghahramani, 2016). More importantly, deep learning systems with more reliable predictive probability can provide better feedback for explaining what is going on, situations when its prediction becomes uncertain, and unexpected anomalies to users. Unfortunately, neural networks are prone to be overconfident and lack uncertainty representation ability, and this problem has become a fundamental concern in the deep learning community.

Bayesian methods have *innate abilities* to produce reliable predictive probability. Specifically, they express the probability distribution over parameters, in which uncertainty in the parameter space is automatically determined by data (MacKay, 1992; Neal, 1993). Then, uncertainty in prediction can be represented by means of providing rich information about aggregated predictions from different parameter configurations such as entropy and mutual information. From this perspective, *deterministic* neural networks selecting a single parameter configuration that cannot provide such rich information naturally lack the uncertainty representation ability. However, the automatic determination of parameter uncertainty in the light of data, i.e., *posterior inference*, comes with prohibitive computational costs. Therefore, the mainstream approach for improving the predictive probability quality has been an efficient adoption of the Bayesian principle into neural networks (Gal & Ghahramani, 2016; Ritter et al., 2018; Teye et al., 2018; Joo et al., 2020a).

Recent works (Lakshminarayanan et al., 2017; Müller et al., 2019; Thulasidasan et al., 2019) has discovered *the hidden gems* of label smoothing (Szegedy et al., 2016), mixup (Zhang et al., 2018), and

adversarial training (Goodfellow et al., 2015), which improve the calibration performance and the uncertainty representation ability. These findings present a new possibility of improving the reliability of the predictive probability without changing the deterministic nature of neural networks. This direction is appealing because it can be applied in a plug-and-play fashion to the existing building blocks. This means that they can *inherit* the scalability, computational efficiency, and surprising generalization performance of the deterministic neural networks, for which Bayesian neural networks often struggle (Wu et al., 2019; Osawa et al., 2019; Joo et al., 2020a).

Motivated by these observations, we investigate a general direction from *the regularization perspective* to mitigate the unreliable predictive probability problem, rather than proposing new constructive heuristics or discovering hidden properties of specific methods. Our main contribution is twofold. First, we present a new direction for alleviating the unreliable predictive behavior, which is readily applicable, computationally efficient, and scalable to large-scale models compared to Bayesian neural networks or ensemble methods. Second, our findings provide a novel view of the role of explicit regularization in deep learning, which improves the reliability of the predictive probability.

## 2 ANALYZING THE CAUSE OF UNRELIABLE PREDICTIVE PROBABILITY

### 2.1 BACKGROUND

We consider a classification problem with i.i.d. training samples $\mathcal{D} = \left\{ (\boldsymbol{x}^{(i)}, y^{(i)}) \right\}_{i=1}^{N}$ drawn from unknown distributions $P_{\mathbf{x},\mathbf{y}}$ whose corresponding tuple of random variables is $(\mathbf{x}, \mathbf{y})$. We denote $\mathcal{X}$ as an input space and $\mathcal{Y}$ as a set of categories $\{1, 2, \cdots, K\}$. Let $f^{\boldsymbol{W}} : \mathcal{X} \to \mathcal{Z}$ be a neural network with parameters $\boldsymbol{W}$ where $\mathcal{Z} = \mathbb{R}^{K}$ is a logit space. On top of the logit space, the softmax $\sigma : \mathbb{R}^{K} \to \triangle^{K-1}$ normalizes the exponential of logits:

$$\phi_k^{\boldsymbol{W}}(\boldsymbol{x}) = \frac{\exp(f_k^{\boldsymbol{W}}(\boldsymbol{x}))}{\sum_i \exp(f_i^{\boldsymbol{W}}(\boldsymbol{x}))} \tag{1}$$

where we let $\phi_k^{\boldsymbol{W}}(\boldsymbol{x}) = \sigma_k(f^{\boldsymbol{W}}(\boldsymbol{x}))$ for brevity. $\sigma_k(f^{\boldsymbol{W}}(\boldsymbol{x}))$ is often interpreted as the predictive probability that the label of $\boldsymbol{x}$ belongs to class $k$ (Bridle, 1990)

The probabilistic interpretation of neural network outputs gives the natural minimization objective for classification–the cross-entropy between the predictive probability and the one-hot encoded label: $l_{CE}(\mathbf{y}, \phi^{\boldsymbol{W}}(\mathbf{x})) = -\sum_k \mathbb{1}_{\mathbf{y}}(k) \log \phi_k^{\boldsymbol{W}}(\mathbf{x})$, where $\mathbb{1}_{\mathcal{A}}(\omega)$ is an indicator function taking one if $\omega \in \mathcal{A}$ and zero otherwise. By minimizing the cross-entropy (or equivalently maximizing the log-likelihood) with stochastic gradient descent (SGD) (Robbins & Monro, 1951) or its variants, modern neural networks achieve the surprising generalization performance.

As the demand for neural networks in real-world decision-making is emerging, reliable predictive probability has been of interest in the machine learning community. One important quality of predictive probability is calibrated behavior. Specifically, based on the notion of calibration in classical forecasting problem (Dawid, 1982), the perfectly calibrated model can be defined as follows:

$$p(y = k|\phi^{\boldsymbol{W}}(\mathbf{x}) = \boldsymbol{p}) = \boldsymbol{p}_k, \quad \forall \boldsymbol{p} \in \triangle^{K-1}, k \in \{1, 2, \cdots, K\} \tag{2}$$

Here note that the calibrated model does not necessarily be ones producing $\phi_k^{\boldsymbol{W}}(\mathbf{x}) = p(y = k|\mathbf{x})$.

In practice, expected calibration error (ECE) (Naeini et al., 2015) is widely used for calibration performance measure. ECE on dataset $\mathcal{D}^T$ can be computed by binning predictions into $M$-groups based on their confidences[1] and then averaging their calibration scores by:

$$\sum_{i=1}^{M} \frac{|\mathcal{G}_i|}{|\mathcal{D}^T|} |\text{acc}(\mathcal{G}_i) - \text{conf}(\mathcal{G}_i)| \tag{3}$$

where $\mathcal{G}_i = \left\{ \boldsymbol{x} : i/M < \max_k \phi_k^{\boldsymbol{W}}(\boldsymbol{x}) \leq (1+i)/M, \boldsymbol{x} \in \mathcal{D}^T \right\}$; $\text{acc}(\mathcal{G}_i)$ and $\text{conf}(\mathcal{G}_i)$ are average accuracy and confidence of predictions in group $\mathcal{G}_i$, respectively.

---

[1]Throughout this paper, the confidence (or predictive confidence) at $\boldsymbol{x}$ refers to $\max_k \phi_k^{\boldsymbol{W}}(\boldsymbol{x})$, which is different from the confidence in statistics literature.

Another metric for evaluating the reliability of predictive probability is based on predictive entropy, which evaluates how well is the model aware of its ignorance. To this end, predictive entropy is measured on samples that the model is ignorant of, such as misclassified or out-of-distribution (OOD) samples. For the classifier having reliable predictive probability, we expect the high uncertainty of $\phi^{\boldsymbol{W}}(\mathbf{x})$ on such samples, i.e., the answer "I don't know."

However, several recent findings show that the resulting neural network produces unreliable predictive probability, which interprets the softmax output as the "predictive probability" implausible (Gal & Ghahramani, 2016). For example, Figure 1 illustrates the unreliable predictive behavior of ResNet (He et al., 2016): the network produces outputs with high confidence in misclassified examples (Figure 1 (upper)) and provides low predictive entropy on out-of-distribution samples, albeit the samples belong to none of the classes it learned (Figure 1 (lower)).

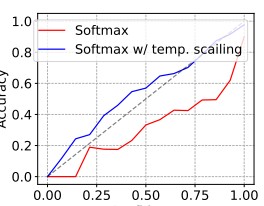

One simple yet effective solution for improving the predictive probability's quality is temperature scaling (Guo et al., 2017), which adjusts the smoothness of the softmax so that the resulting predictive probability maximizes the log-likelihood on unseen dataset $\mathcal{D}'$:

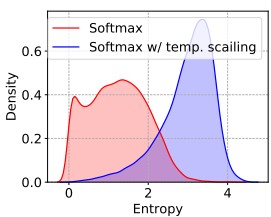

$$\max_{\tau} \sum_{(\boldsymbol{x},y)\in\mathcal{D}'} \log \frac{\exp(f_y^{\boldsymbol{W}}(\boldsymbol{x})/\tau)}{\sum_j \exp(f_j^{\boldsymbol{W}}(\boldsymbol{x})/\tau)} \qquad (4)$$

where $\boldsymbol{W}$ is a fixed pretrained weight and $\tau$ is a temperature controlling the smoothness of the softmax output. This simple method makes the softmax output more *reliable predictive probability*. For instance, the predictive confidence matches its actual accuracy well, and the predictive entropy on out-of-distribution samples significantly increases (Figure 1).

Figure 1: Reliability curve (upper) and predictive uncertainty on out-of-distribution samples (lower) of ResNet.

## 2.2 A CLOSER LOOK AT THE LOG-LIKELIHOOD ON UNSEEN SAMPLES.

Motivated by the success of the temperature scaling, we aim to find the relationship between the log-likelihood and the calibration performance. To this end, let s be a binary random variable indicating whether a model correctly classifies a sample. Then, we can derive the following upper bound by using the law of total expectation:

$$\mathbb{E}_{\mathbf{x},\mathbf{y}}[\log \phi_{\mathbf{y}}^{\boldsymbol{W}}(\mathbf{x})] = \mathbb{E}_{\mathbf{x}} \left[ p_{s|\mathbf{x}}(s=1)\mathbb{E}_{\mathbf{y}|s=1,\mathbf{x}} \left[\log \phi_{\mathbf{y}}^{\boldsymbol{W}}(\mathbf{x})\right] + p_{s|\mathbf{x}}(s=0)\mathbb{E}_{\mathbf{y}|s=0,\mathbf{x}} \left[\log \phi_{\mathbf{y}}^{\boldsymbol{W}}(\mathbf{x})\right]\right]$$

$$\leq \mathbb{E}_{\mathbf{x}} \left[ p_{s|\mathbf{x}}(s=1) \log \phi_{m_{\mathbf{x}}}^{\boldsymbol{W}}(\mathbf{x}) + p_{s|\mathbf{x}}(s=0) \log(1 - \phi_{m_{\mathbf{x}}}^{\boldsymbol{W}}(\mathbf{x}))\right] \quad (5)$$

where $m_{\mathbf{x}}$ is the predictive class such that $m_{\mathbf{x}} = \arg\max_k f_k^{\boldsymbol{W}}(\mathbf{x})$, $p_{s|\mathbf{x}}(s=1) = p(y = \arg\max_k \phi_{m_k}^{\boldsymbol{W}}(\mathbf{x})|\mathbf{x})$, and the inequality comes from the fact that $\mathbb{E}_{\mathbf{y}|s=0,\mathbf{x}} \left[\log \phi_{\mathbf{y}}^{\boldsymbol{W}}(\mathbf{x})\right] \leq \max_{k\neq m_{\mathbf{x}}} \log \phi_k^{\boldsymbol{W}}(\mathbf{x}) \leq \log(1 - \phi_{m_{\mathbf{x}}}^{\boldsymbol{W}}(\mathbf{x}))$. The upper bound can be thought of as the probabilistic measure of the calibration performance. Specifically, suppose a neural network produces an answer with probability $\phi_{m_{\mathbf{x}}}^{\boldsymbol{W}}(\mathbf{x})$ and refuses to answer otherwise. Then, the upper bound becomes the divergence between the model's ability to correctly predict the sample and the model's willingness to answer. When we consider the inequality in equation 5 is applied to the empirical mean on the test dataset, this inequality clearly explains why the temperature scaling is helpful; it increases a lower bound of the calibration error with modified confidence by $\tau$ closer to its accuracy.

More importantly, the inequality in equation 5 explains the impacts of the cross-entropy minimization on the behavior of the neural network. Specifically, suppose $\mathbb{E}_{(\mathbf{x},\mathbf{y})\sim\mathcal{D}}[\log \phi_{\mathbf{y}}^{\boldsymbol{W}}(\mathbf{x})] \to 0$ for some dataset $\mathcal{D}$. Then, it can be shown that $p_{s|\mathbf{x}}(s=1) \to 1$ and $\phi_{m_{\boldsymbol{x}}}^{\boldsymbol{W}}(\boldsymbol{x}) \to 1$ for all $(\boldsymbol{x},y) \in \mathcal{D}$. This holds because we have $\min_k \phi_k^{\boldsymbol{W}}(\boldsymbol{x}) \geq 1/K$ for any $\boldsymbol{x}$ and the minimum of $p \log q + (1 - p) \log(1 - q)$ are $(p,q) \to (1,1)$ or $(p,q) \to (0,0)$. Therefore, for high-capacity models that can make the log-likelihood on the training dataset $\mathcal{D}_{tr}$ close to zero, e.g., deep neural networks, its behavior on the training set will converge to a configuration that corrects all samples *with perfect confidence*. Figure 2 illustrates this phenomenon in ResNet trained on CIFAR-100: $\mathbb{E}_{\mathbf{x}\sim\mathcal{D}_{tr}}[\log \phi_{m_{\mathbf{x}}}^{\boldsymbol{W}}] \to 0$ (a) and $\mathbb{E}_{(\mathbf{x},\mathbf{y})\sim\mathcal{D}_{tr}}[p_{s|\mathbf{x}}(s=1)] \to 1$ (b) as the training continues.

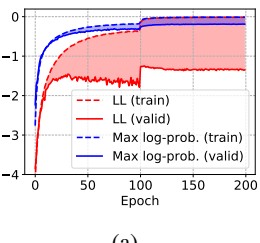 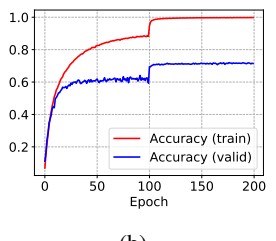 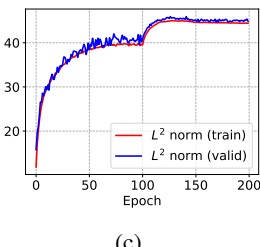

(a)                              (b)                              (c)

Figure 2: Monitoring changes in the behavior of ResNet during training on CIFAR-100. In (a), LL indicates the log-likelihood $\mathbb{E}_{\mathbf{x},\mathbf{y}}[\log \phi_{\mathbf{y}}^{\boldsymbol{W}}(\mathbf{x})]$ and max log-prob indicates $\mathbb{E}_{\mathbf{x}}[\log \phi_{m_{\mathbf{x}}}^{\boldsymbol{W}}(\mathbf{x})]$. In (c), the $L^2$ norm is approximated with respect to empirical distributions of training samples and validation samples.

Why are these convergences problematic? We observe that function's properties, which depends only on $\mathbf{x}$, evaluated on two different datasets are very close to each other if they are drawn from the same distribution, unlike values that depend on the external randomness $\mathbf{y}|\mathbf{x}$; that is, $|\mathbb{E}_{\mathbf{x}\sim\mathcal{D}_{tr}}[g(\mathbf{x})]-\mathbb{E}_{\mathbf{x}'\sim\mathcal{D}_{val}}[g(\mathbf{x}')]| \ll |\mathbb{E}_{(\mathbf{x},\mathbf{y})\sim\mathcal{D}_{tr}}[h(\mathbf{x},\mathbf{y})] - \mathbb{E}_{(\mathbf{x}',\mathbf{y}')\sim\mathcal{D}_{val}}[h(\mathbf{x}',\mathbf{y}')]|$ for some functions $g(\cdot)$ and $h(\cdot)$. For example, the empirical mean of the maximum log-probability $\log \phi_{m_{\mathbf{x}}}^{\boldsymbol{W}}(\mathbf{x})$ (Figure 2 (a)) and $L^2$ norm of $f^{\boldsymbol{W}}$ (Figure 2 (c))$^2$ on training samples are significantly similar to those values on unseen samples compared to the log-likelihood (Figure 2 (a)) and the accuracy (Figure 2 (b)). We conjecture that the log-likelihood maximization with a high-capacity neural network naturally results in a high calibration error on unseen samples due to this discrepancy; that is, it produces perfect confidence on unseen samples as it does on the training samples where it cannot produce the perfect accuracy.

Fortunately, this result also indicates that restricting the predictive confidence $\phi_{m_{\mathbf{x}}}^{\boldsymbol{W}}(\mathbf{x})$ on training samples will directly impact the predictive confidence on unseen samples and therefore can reduce the calibration error. For this reason, we explore various ways to restrict confidence and show their efficacy on a wide range of tasks for the rest of the paper.

## 3 CONFIDENCE CONTROL FOR RELIABLE PREDICTIVE PROBABILITY

The goal of this section is to examine the effectiveness of restricting predictive confidence on improving the reliability of predictive probability. Therefore, we apply several existing regularizers to the (pre-activation) ResNet (He et al., 2016) trained on CIFAR (Krizhevsky et al., 2009), which is one of the most prevalent basis models in many state-of-the-art architectures (Huang et al., 2017; Xie et al., 2017). We also present the VGG (Simonyan & Zisserman, 2015) as a representative of models without residual connection in appendix B, in which we observe results similar to ResNet. We follow the standard training strategy presented in He et al. (2016) except for the initial learning rate warm-up, clipping the gradient when its norm exceeds one, and making an extra validation set of 10,000 samples split from the training set. We describe a detailed setup in appendix A.

We mainly use the ECE for measuring the reliability of the predictive probability. In addition, we used negative log-likelihood (NLL), following the common practice (Lakshminarayanan et al., 2017; Ovadia et al., 2019; Guo et al., 2017). Here, NLL evaluates how well the predictive probability explains the test data $\mathcal{D}^T$, whose optimal score is achieved if and only if $\phi^{\boldsymbol{W}}(\mathbf{x})$ perfectly matches $p(\mathbf{y}|\mathbf{x})$.

### 3.1 CONFIDENCE CONTROL BY WEIGHT DECAY

The simplest way to constrain the confidence may conceivably be using stronger weight decay (Krogh & Hertz, 1992), which can encourage the production of less extreme outputs by shrinking weights. Therefore, we first explore the confidence control by varying the weight decay ratio $\lambda$, conjecturing that the weight decay ratio used in the base model, e.g., $\lambda = 0.0001$ in ResNet,

---

$^2$This norm is evaluated on the function space by $\| f^{\boldsymbol{W}} \|_2 = \left(\int |f^{\boldsymbol{W}}(\boldsymbol{x})|^2 dP_{\mathbf{x}}(\boldsymbol{x})\right)^{1/2}$, which we will discuss with more detail in Section 3.2.

is too small to prevent overconfident predictions. Figure 3 (upper) illustrates the impact of $\lambda$ on generalization performance and calibration performance. We can see that ECE decreases to some extent as the decay ratio increases. However, the ECE improvement becomes *inversely* proportional to a generalization performance improvement when the decay ratio is larger than 0.001. This means that improving the reliability with strong weight decay is against the primary goal of supervised learning.

We further investigate this undesirability by monitoring the $L^2$ function norm of $f^{\boldsymbol{W}}$ (cf. equation 7) under different weight decay ratios. This analysis is motivated by the fact that temperature scaling modifies only the $L^p$ norm of $f^{\boldsymbol{W}}$ as it divides logits with a scalar. Under the single-label classification model, this measure is a useful proxy for the predominance of the maximum predictive probability. To set the desired value of the $L^2$ norm, we "leak" the test set $\mathcal{D}^T$ and then apply temperature scaling to the neural network trained with $\lambda = 0.0001$.

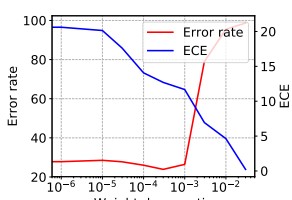

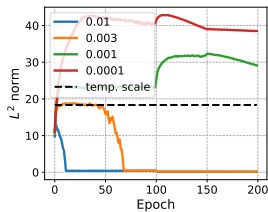

Figure 3 shows that the SGD with various decay ratios finds only *a trivial solution* or *an infeasible solution* to the following optimization problem:

$$\min_{\boldsymbol{W}} \hat{\mathbb{E}}_{\mathcal{D}} \left[ l_{CE}(\mathbf{y}, \phi^{\boldsymbol{W}}(\mathbf{x})) \right] \quad \text{s.t. } \parallel f^{\boldsymbol{W}} \parallel_2 \leq \parallel f^{\boldsymbol{W}'}/\tau^* \parallel_2 \quad (6)$$

where $\hat{\mathbb{E}}_{\mathcal{D}}[\cdot]$ is the empirical mean on $\mathcal{D}$, $\boldsymbol{W}'$ is the trained weights under $\lambda = 0.0001$, and $\tau^* = \arg\max_\tau \hat{\mathbb{E}}_{\mathcal{D}^T} \left[ \log \left\{ \sigma(f_\mathbf{y}^{\boldsymbol{W}'}(\mathbf{x})/\tau) \right\} \right]$. This means that reducing the predictive confidence to the desired level without sacrificing generalization performance is a significantly challenging optimization problem. Specifically, Figure 3 (lower) shows that

Figure 3: Impact of the weight decay ratio on ECE and accuracy (upper) and the $L^2$ norm (lower).

the $L^2$ norm goes to zero under the decay ratio $\lambda \geq 0.003$, which means that all weights collapse to zero, i.e., the trivial solution. This happens when the magnitude of the decay ratio overwhelms the magnitude of the weight update by the gradient of the cross-entropy, e.g., at approximately the epoch 50 under $\lambda = 0.003$ (Figure 3 (lower)). However, ratios of 0.001 and 0.0001 do not suffer from the weight collapse, but the scale of the $L^2$ norm under such ratios is significantly higher than $\parallel f^{\boldsymbol{W}'}/\tau^* \parallel_2$, which corresponds to infeasible solutions (Figure 3 (lower)). These results may seem natural because the weight decay does not explicitly consider the constraint about predictive confidence. Therefore, we explore an alternative method that adds a regularization loss explicitly concerning predictive confidence, e.g., the constraint in equation 6.

## 3.2 DIRECT CONFIDENCE CONTROL BY EXPLICIT REGULARIZATION

In this subsection, we examine two types of regularizers that directly constrain the predictive confidence on training samples.

**Regularization in the function space.** The first approach regards $f^{\boldsymbol{W}}$ as an element of $L^p(\mathcal{X})$ space. $L^p$ space is the space of measurable functions with the norm:

$$\parallel f^{\boldsymbol{W}} \parallel_p = \left( \int_{\mathcal{X}} |f^{\boldsymbol{W}}(\boldsymbol{x})|^p dP_\mathbf{x}(\boldsymbol{x}) \right)^{1/p} < \infty \quad (7)$$

Here, we note that the norm is computed with respect to the input generating distribution $P_\mathbf{x}$, which allows concerning how the function $f^{\boldsymbol{W}}$ actually behaves on the data manifold. Since $P_\mathbf{x}$ is unknown, so it is approximated by the Monte-Carlo approximation with minibatch samples. Then, the approximate function norm can be computed by $\parallel f^{\boldsymbol{W}} \parallel_p^p \approx \frac{1}{m} \sum_{i,j} |f_j^{\boldsymbol{W}}(\boldsymbol{x}^{(i)})|^p$. By penalizing the complexity with the $L^p$ norm, the continuous increase in the leading entry of logit towards infinity, or continuous decreases in the nonleading entry to negative infinity, can be prevented. In this paper, we examine $\parallel f^{\boldsymbol{W}} \parallel_1$ and $\parallel f^{\boldsymbol{W}} \parallel_2^2$ regularization losses.

**Regularization in the probability distribution space.** The second approach regards $f^{\boldsymbol{W}}(\mathbf{x})$ as a random variable and then minimize its distance to the bell-shaped target distribution. Since the

Table 1: Experimental results under various regularization methods on ResNet. Arrows on the metrics represent the desirable direction. We searched four hyperparameters for each method and chose the best hyperparameter based on validation accuracy (cf. appendix A). Values represent mean obtained from five repetitions, and all values are rounded to two decimal places.

| method | CIFAR-10 | | | | CIFAR-100 | | | |
|---|---|---|---|---|---|---|---|---|
| | Acc $\uparrow$ | NLL $\downarrow$ | ECE $\downarrow$ | $\parallel f^{\boldsymbol{W}} \parallel_2$ | Acc $\uparrow$ | NLL $\downarrow$ | ECE $\downarrow$ | $\parallel f^{\boldsymbol{W}} \parallel_2$ |
| Vanilla | 94.17 | 0.29 | 4.06 | 20.12 | 74.64 | 1.31 | 13.95 | 43.81 |
| $\parallel f^{\boldsymbol{W}} \parallel_1$ | 94.32 | 0.25 | 2.89 | 6.81 | 76.28 | 1.27 | 7.77 | 9.07 |
| $\parallel f^{\boldsymbol{W}} \parallel_2^2$ | 94.38 | 0.23 | 3.27 | 8.19 | 75.84 | 1.07 | 5.52 | 10.55 |
| $SW_1(\mu_{\mathcal{D}'}^{\boldsymbol{W}}, \nu)$ | 94.46 | 0.23 | 2.12 | 5.4 | 76.27 | 1.1 | 7.02 | 9.67 |
| PER | 94.30 | 0.23 | 2.89 | 6.94 | 76.23 | 1.15 | 4.67 | 7.56 |

bell-shaped distribution has a peak mass at zero and the same density on points with the same distance to zero, this regularization can make logits have small values and make each entry of logits have a similar scale, which can prevent neural nets from producing overconfident answers. Among various bell-shaped distributions, we choose the standard Gaussian for the target distribution due to its simplicity and popularity. As a metric in the probability distribution space, we use the sliced Wasserstein distance of order one because of its computational efficiency and ability to measure the distance between probability distributions with different supports, which is useful when dealing with the empirical distribution. We refer to Peyré et al. (2019) for more detailed explanations about this metric. Specifically, given minibatch samples $\mathcal{D}' = \{\boldsymbol{x}^{(i)}\}_{i=1}^{m}$, let an empirical measure of logits be $\mu_{\mathcal{D}'}^{\boldsymbol{W}}(\mathbb{A}) = \frac{1}{m} \sum_i \mathbb{1}_{\mathbb{A}}(f^{\boldsymbol{W}}(\boldsymbol{x}^{(i)}))$ and the standard Gaussian measure on $\mathcal{Z}$ be $\nu(\mathbb{A}) = \frac{1}{2\pi^{K/2}} \int_{\mathbb{A}} \exp\left(-\frac{1}{2} \parallel \boldsymbol{z} \parallel^2\right) d\boldsymbol{z}$. Then, the sliced Wasserstein distance can be computed by:

$$SW_1(\mu_{\mathcal{D}'}^{\boldsymbol{W}}, \nu) = \int_{\mathbb{S}^{K-1}} \int_{-\infty}^{\infty} \left| F_{\mu_\theta}(x) - \frac{1}{m} \sum_{i=1}^{m} \mathbb{1}_{(\infty, x)}(\langle \boldsymbol{z}^{(i)}, \theta \rangle) \right| dx d\lambda(\theta) \tag{8}$$

where $\boldsymbol{z}^{(i)} = f^{\boldsymbol{W}}(\boldsymbol{x}^{(i)})$, $\lambda$ is a uniform measure on the unit sphere $\mathbb{S}^{K-1}$, and $\mu_\theta$ is a measure obtained by projecting $\mu_{\mathcal{D}'}^{\boldsymbol{W}}$ at angle $\theta$. Projected error function regularization (PER) (Joo et al., 2020b) simplifies the computation of the $SW_1(\mu_{\mathcal{D}'}^{\boldsymbol{W}}, \nu)$ by applying the Minkowski inequality to the above equation. As a result, the gradient of PER resembles the gradient of Huber loss (Huber, 1964) in the projected space, which allows combining advantages of both the $L^1$ norm and the $L^2$ norm as well as capturing the dependency between logits of each location by a projection operation (Joo et al., 2020b).

**Image classification with ResNet.** Table 1 lists the experimental results, in which both regularization in the function space and the Wasserstein probability space successfully controlled the confidence, e.g., reduced the $L^2$ norm of ResNet by at least 34% on CIFAR-10 and 68% on CIFAR-100. We note that regularization methods can constrain the confidence without compromising the generalization performance; actually, all regularization methods achieves small but consistent improvements of test error rates. We also note that the sum of the Frobenius norm of weights often increases compared to the vanilla method and changes only at most 2% when it decreases, which again shows the undesirability of adjusting the weight decay ratio for confidence control.

More importantly, the predictive probability's reliability significantly improves under all considered measures compared to the vanilla method. For instance, the regularization methods reduce the NLL of ResNet by at least 13% CIFAR-10 and 6% on CIFAR-100 and reduce the ECE of ResNet by at least 19% on CIFAR-10 and 41% on CIFAR-100. These improvements are comparable to or better than those of temperature scaling. For instance, ResNet with temperature scaling gives an NLL of 1.15 and an ECE of 8.41 on CIFAR-100. Here, we split the test set into two equal-size sets–a performance measurement set and a temperature calibration set–and measure the performance after temperature scaling with the calibration set, and repeat the same procedure by reversing their roles. We note that a more realistic evaluation requires drawing the temperature calibration set from the training set. In this case, its performance decreases as it cannot fully exploit the entire dataset during training.

**Predictive uncertainty.** We also investigate the usefulness of explicit regularization on the uncertainty representation ability on misclassified samples and OOD samples. Since the model is ignorant

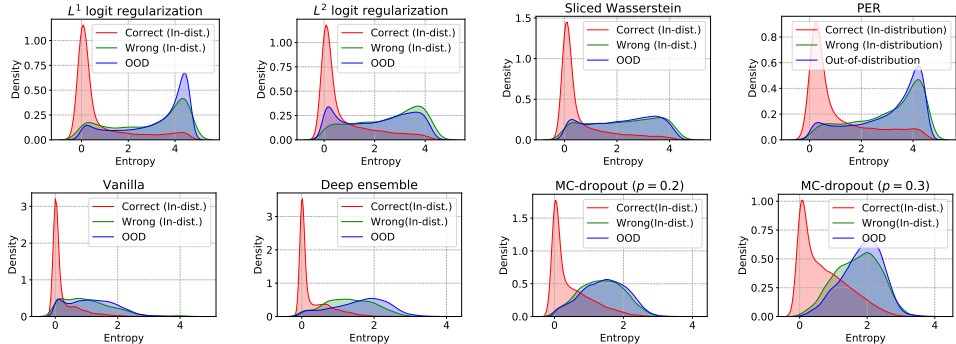

Figure 4: Density of predictive uncertainty on CIFAR-100 (in-distribution) and SVHN (OOD). The upper figures illustrate explicit regularization methods, and the lower figures illustrate the vanilla method, ensemble methods, and Bayesian neural networks.

Table 2: Misclassification detection and OOD detection task performances based on NBAUCC$_{0.5}$ (higher is better). MC-D ($p$) represents MC-dropout with probability $p$, and Ens represents deep ensemble.

| Task | Vanilla | MC-D (0.2) | MC-D (0.3) | Ens. | $\| f^{\boldsymbol{W}} \|_1$ | $\| f^{\boldsymbol{W}} \|_2^2$ | $SW_1(\mu_{\mathcal{D}'}^{\boldsymbol{W}}, \nu)$ | PER |
|------|---------|-----------|-----------|------|---------|---------|-----------|-----|
| Miscls | 1.55 | 3.33 | 5.94 | 0.59 | 9.53 | 6.85 | 5.10 | 10.24 |
| OOD | 15.98 | 22.14 | 31.28 | 28.46 | 55.51 | 31.47 | 31.63 | 49.77 |

of misclassified samples and OOD samples as they do not belong to any category, the neural network should produce the answer of "I don't know" for these samples. Figure 4 illustrates the predictive uncertainty of ResNet-50 with respect to CIFAR-100 (in-distribution) and SVHN (Netzer et al., 2011) (OOD). The vanilla method's predictive uncertainty on SVHN and misclassified samples remains in the somewhat confident region, albeit less confident compared to those on CIFAR-100. In contrast, explicit regularization successfully gathers a mass of predictive uncertainty for both SVHN samples and misclassified samples around the maximum-entropy region.

We compare the uncertainty representation abilities of regularization methods to those of Bayesian neural networks and ensemble methods. Specifically, we use the scalable Bayesian neural network, called MC-dropout (Gal & Ghahramani, 2016), because other methods based on variational inference (Graves, 2011; Blundell et al., 2015; Wu et al., 2019) or MCMC (Welling & Teh, 2011; Zhang et al., 2020) require modifications to the baseline including the optimization procedure and the architecture, which deters a fair comparison. We searched a dropout rate over {0.1, 0.2, 0.3, 0.4, 0.5 } and used 100 number of Monte-Carlo samples at test time, i.e., 100x more inference time. We also use the deep ensemble (Lakshminarayanan et al., 2017) with 5 number of ensembles, i.e., 5x more training and inference time. Figure 4 shows that the regularization-based methods produce significantly better uncertainty representation than the MC-dropout and deep ensemble; even though both deep ensemble and MC-dropout can move mass on less certain regions, the positions are still far from the highest uncertainty region, unlike the regularization-based methods. Finally, we note several approaches (Malinin & Gales, 2019; Papadopoulos et al., 2019) explicitly maximize uncertainty on OOD training samples for OOD detection tasks. However, we do not consider such approaches, assuming we have no access to OOD samples during training.

**Misclassification/OOD detection.** We also evaluate the uncertainty representation ability for misclassified and out-of-distribution samples. To this end, we use recently proposed normalized bounded area under the calibration curve (NBAUCC) (Kong et al., 2020), which handles the disadvantage of classical measures such as AUROC and AUPRC that cannot take the calibration property into account. Specifically, NBAUCC with the upper bound of confidence threshold for misclassified

Table 3: Experimental results under various regularization methods on BERT. Arrows on the metrics represent the desirable direction. Values represent mean obtained from five repetitions, which are rounded to two decimal places.

| method | 20-Newsgroup | | | | Web of science | | | |
|---|---|---|---|---|---|---|---|---|
| | Acc ↑ | NLL ↓ | ECE ↓ | $\| f^{\boldsymbol{W}} \|_2$ | Acc ↑ | NLL ↓ | ECE ↓ | $\| f^{\boldsymbol{W}} \|_2$ |
| Vanilla | 84.51 | 0.70 | 6.40 | 9.71 | 81.28 | 0.94 | 8.26 | 20.13 |
| $\| f^{\boldsymbol{W}} \|_1$ | 84.71 | 0.61 | 4.67 | 6.02 | 81.68 | 0.90 | 7.02 | 8.89 |
| $\| f^{\boldsymbol{W}} \|_2^2$ | 85.04 | 0.64 | 3.24 | 5.11 | 81.83 | 0.94 | 5.79 | 7.70 |
| $SW_1(\mu_{\mathcal{D}'}^{\boldsymbol{W}}, \nu)$ | 84.76 | 0.67 | 3.78 | 4.98 | 81.44 | 0.89 | 7.22 | 11.33 |
| PER | 85.02 | 0.65 | 4.50 | 6.21 | 81.36 | 0.87 | 8.19 | 13.21 |

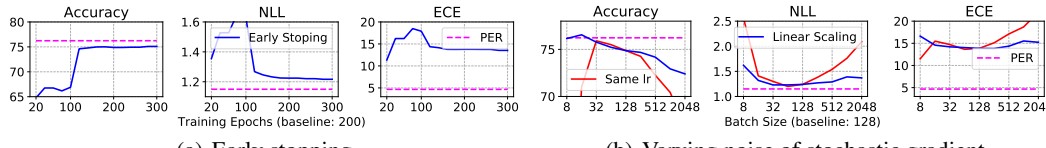

(a) Early stopping                    (b) Varying noise of stochastic gradient

Figure 5: Effects of implicit regularization methods on accuracy, negative log-likelihood, and ECE.

or OOD samples, denoted by $\tau$, is computed by:

$$\text{NBAUCC}_\tau = \frac{1}{M} \sum_{i=1}^{M} F_1 \left( \frac{\tau}{M} i \right) \tag{9}$$

where $F_1(t)$ computes $F_1$ score by regarding samples with predictive confidence higher than $\tau$ as correct (resp. in-distribution) samples and incorrect (resp. OOD) samples otherwise; $M$ is a predetermined step size. We present the misclassification/OOD detection performance in Table 2, where all regularization methods significantly improve detection performances based on their better predictive uncertainty representation abilities.

**Document classification with BERT.** Finally, we perform additional experiments on the natural language processing domain in order to more thoroughly evaluate the effectiveness of explicit regularization. Inspired by the recent finding (Kong et al., 2020), we perform document classification on 20 newsgroup dataset (Socher et al., 2012) and web of science dataset (Kowsari et al., 2017). We obtain a classifier by adding one linear layer on top of the BERT (Devlin et al., 2018). We provide training details in Appendix A. As consistent with the findings in the image classification task, all explicit regularization methods improve the reliability of predictive probability with consistent gains in test accuracy (Table 3).

### 3.3 CAN THE IMPROVED RELIABILITY BE ATTRIBUTED TO IMPROVED GENERALIZATION?

Considering that the log-likelihood is the training objective, the improved reliability may be caused by the improved generalization *due to regularization effects*. To clarify this point, we investigate the impact of improved generalization via implicit regularization mechanisms (early stopping and exploitation of noise from SGD) to ResNet-50 on CIFAR-100. Figure 5 (a) shows that the early stopping does not result in dramatic improvements in all measures compared to PER. Instead, NLL and ECE slightly improve under longer training due to increased classification accuracy. Considering that the $L^2$ norm of logits achieves a significantly high value at the early stage of training (cf. Figure 2 (b)), this undesirable result may seem natural. Next, we examine the effects of varying batch sizes under the baseline learning rate or linear scaling rule (Goyal et al., 2017), in which a smaller batch size increases the noise of the stochastic gradient. In Figure 5 (b), we observe that increasing the noise level by using smaller batch sizes benefits generalization performance. However, it does not benefit and even worsens the predictive distribution quality (NLL and ECE). In addition, we observed that these implicit regularizers fail to produce high predictive entropy on OOD samples, unlike explicit regularization. These observations corroborate the efficacy of confidence control in improving the reliability of predictive probability, which highlights the importance of explicit regularization.

## 4 RELATED WORK

Recent works show that joint modeling of a generative model $p(\mathbf{x})$ along with a classifier $p(\mathbf{y}|\mathbf{x})$, or $p(\mathbf{x}, \mathbf{y})$ directly, helps to produce reliable predictive probability (Alemi et al., 2018; Nalisnick et al., 2019; Grathwohl et al., 2020). Specifically, Alemi et al. (2018) argue that modeling stochastic hidden representation through the variational information bottleneck principle (Alemi et al., 2017) allows representing better predictive uncertainty. This can be related to the effectiveness of ensemble methods, which aggregate representations of several models, on enhancing predictive uncertainty representation and confidence calibration (Lakshminarayanan et al., 2017; Ovadia et al., 2019; Ashukha et al., 2020). In this regard, hybrid modeling and ensemble methods share a similar principle to the Bayesian methods, concerning the *stochasticity of the function*. However, this paper concentrates on explicit regularization for controlling the predictive confidence, which is fundamentally different from previous works focusing on building the stochastic representation.

Other works concentrate on *the structural characteristics* of neural networks. Specifically, Hein et al. (2019) identify the cause of the overconfidence problem based on an analysis of the affine compositional function, e.g., ReLU. The basic intuition behind this analysis is that one can always find a multiplier $\lambda$ to an input $\boldsymbol{x}$, which makes a neural network produce one dominant entry on $\lambda\boldsymbol{x}$. Verma & Swami (2019) point out that the region of the highest predictive uncertainty under the softmax forms a subspace in the logit space, so the volume of an area representing high predictive uncertainty would be negligible. However, our approach suggests that these structural characteristics' inherent flaws can be easily cured by adding an explicit regularization term without changing the existing components of neural networks.

From the perspective of the statistical learning theory (Vapnik, 1995), a regularization method minimizing some form of complexity measures, e.g., Rademacher complexity (Bartlett et al., 2005) or VC-dimension (Vapnik & Chervonenkis, 2015), is a "must" to achieve better generalization of overparameterized models by preventing memorization of intricate patterns existing only in training samples. However, the role of capacity control with explicit regularization is challenged by many observations in deep learning. Specifically, overparameterized neural networks achieve impressive generalization performance with only implicit regularizations contained in the optimization procedures (Hardt et al., 2016; Li et al., 2019) or the structural characteristics (Gunasekar et al., 2018; Hanin & Rolnick, 2019; Luo et al., 2019). Moreover, Zhang et al. (2017) show that explicit regularization cannot prevent neural networks from easily fitting random labels that *cannot be generalized* to unseen examples. Therefore, the importance of capacity control with explicit regularization seems to be questionable in deep learning. In this work, we reemphasize its importance, presenting a different view on the role of regularization in terms of *generalization of the predictive probability*, not solely on better accuracy.

## 5 CONCLUSION

In this works, we show the usefulness of the explicit regularization on improving the reliability of predictive probability, which presents a novel view on the role and importance of explicit regularization. Specifically, our extensive experimental results show that several existing regularization methods improves calibration, predictive uncertainty representation, misclassification/OOD detection performances, and even the test accuracy. Compared to the temperature scaling method, the regularization-based approach can be more widely applicable in more general settings such as continual learning and online learning as it does not requires an additional hold-out dataset. In addition, the regularization-based approaches provided a small but consistent gain in generalization performances, while the temperature scaling does not impact the generalization performances. Compared to marginalization-based approach such as Bayesian neural networks and ensemble methods, the regularization-based approach is appealing in terms of computational efficiency and scalability. Despite these advantages, the regularization methods are limited in that they cannot utilize more sophisticated uncertainty quantification methods based on stochastic representation on the predictive probability space, such as mutual information measuring epistemic uncertainty (Smith & Gal, 2018), due to its deterministic nature. We leave this limitation as an important future direction of research, which may be solved by more expressive parameterization, e.g., (Wilson et al., 2016; Skafte et al., 2019; Malinin & Gales, 2019; Joo et al., 2020a).

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

## A    DETAILED EXPERIMENTAL SETUP

**ResNet base setup.** We trained ResNet for 200 epochs by SGD with momentum coefficient 0.9, minibatch size of 128, and a weight decay ratio 0.0001; weights were initialized by He initialization (He et al., 2015); an initial learning rate was 0.1, and decreased by a factor of 10 at 100 and 150 epochs. We also used the standard augmentation presented in He et al. (2015).

**VGG setup.** We trained VGG by re-using the ResNet setup for convenience, except increasing the weight decay ratio to 0.0005 as in Simonyan & Zisserman (2015).

**BERT setup.** Our configuration is based on Kong et al. (2020). Specifically, the model is trained for 5 epochs by Adam optimizer with learning rate 0.00005 and minibatch size of 32.

**Hyperparameters.** We searched four regularization loss coefficients for each method, and chose best one based on validation set accuracy (Table 4). The search spaces were: $\{0.1, 0.03, 0.01, 0.003\}$ for $L^1$ norm; $\{0.03, 0.01, 0.003, 0.001\}$ for $L^2$ norm; $\{0.1, 0.03, 0.01, 0.003\}$ for sliced Wasserstein regularization; $\{1.0, 0.3, 0.1, 0.03\}$ for PER (10x lower coefficient for CIFAR-10).

Sliced Wasserstein regularization and PER involve the integral over the unit sphere, which is evaluated by Monte-Carlo approximation. In this paper, we used 256 number of evaluations, following (Joo et al., 2020b).

Table 4: Best hyperparameters for each configuration

| Regularizer | VGG-16 & CIFAR-10 | VGG-16 & CIFAR-100 | ResNet-50 & CIFAR-10 | ResNet-50 & CIFAR-100 |
|---|---|---|---|---|
| $\parallel f^{\boldsymbol{W}} \parallel_1$ | 0.01 | 0.03 | 0.01 | 0.01 |
| $\parallel f^{\boldsymbol{W}} \parallel_2^2$ | 0.003 | 0.01 | 0.003 | 0.01 |
| $SW_1(\mu_{\mathcal{D}'}^{\boldsymbol{W}}, \nu)$ | 0.001 | 0.03 | 0.001 | 0.01 |
| PER | 0.003 | 1.0 | 0.03 | 1.0 |

## B    VGG RESULTS

As consistent with the results of ResNet, all regularization losses improves NLL, ECE, and accuracy (Table 5), except $L^1$ regularization on CIFAR-100. However, the improvements are less significant compared to ResNet because the small capacity of VGG makes the vanilla method produces less confident answers and then less vulnerable to the confidence penalty. This can be inferred from that values of $\parallel f^{\boldsymbol{W}} \parallel_2$ of VGG are reduced by almost 50% compared to those of ResNet.

Table 5: Experimental results under various regularization methods. Arrows on the metrics represent the desirable direction. Values represent $\mu \pm \sigma$ obtained from five repetitions, and all values are rounded to two decimal places.

| Model & Data | Regularizer | Acc ↑ | NLL ↓ | ECE ↓ | $\parallel f^{\boldsymbol{W}} \parallel_2$ |
|---|---|---|---|---|---|
| VGG-16 & CIFAR-10 | Vanilla | 92.97 ±0.20 | 0.35 ±0.01 | 4.96 ±0.16 | 9.62 ±0.05 |
| | $\parallel f^{\boldsymbol{W}} \parallel_1$ | 93.07 ±0.08 | 0.33 ±0.01 | 4.1 ±0.11 | 6.62 ±0.01 |
| | $\parallel f^{\boldsymbol{W}} \parallel_2^2$ | 93.06 ±0.03 | 0.31 ±0.01 | 4.58 ±0.07 | 7.44 ±0.03 |
| | $SW_1(\mu_{\mathcal{D}'}^{\boldsymbol{W}}, \nu)$ | 93.13 ±0.06 | 0.29 ±0.0 | 1.9 ±0.07 | 5.4 ±0.01 |
| | PER | 93.1 ±0.17 | 0.31 ±0.01 | 4.79 ±0.13 | 8.49 ±0.03 |
| VGG-16 & CIFAR-100 | Vanilla | 71.96 ±0.12 | 1.4 ±0.01 | 16.9 ±0.09 | 22.98 ±0.31 |
| | $\parallel f^{\boldsymbol{W}} \parallel_1$ | 72.71 ±0.17 | 1.44 ±0.01 | 11.37 ±0.18 | 9.27 ±0.1 |
| | $\parallel f^{\boldsymbol{W}} \parallel_2^2$ | 72.68 ±0.16 | 1.31 ±0.01 | 10.79 ±0.28 | 9.28 ±0.11 |
| | $SW_1(\mu_{\mathcal{D}'}^{\boldsymbol{W}}, \nu)$ | 72.26 ±0.23 | 1.35 ±0.01 | 12.59 ±0.34 | 9.72 ±0.02 |
| | PER | 72.89 ±0.24 | 1.34 ±0.01 | 6.47 ±0.12 | 7.37 ±0.05 |

