# OpenReview forum: "Revisiting Explicit Regularization in Neural Networks for Reliable Predictive Probability"
_ICLR.cc/2021/Conference — Reject_

### Official Review · AnonReviewer1 · 2020-10-14
**Proposes an interesting method for learning well-calibrated probability scores in neural networks but is not suitable for publication in its present form.**

**Rating:** 5
**Confidence:** 3

**Review:**

The main contribution of this paper is to propose new regularization methods in deep neural networks that produce well-calibrated probability scores. The authors argue that regularization is better than post-processing, such as temperature scaling, because temperature scaling would require a separate dataset for calibration. In addition, regularization is added to the loss so it does not alter other components of the neural network. There are two forms of regularization that the authors propose: (1) regularizing in the function space, and (2) in the probability space.  Interestingly, they show that both regularization methods yield well-calibrated scores, which cannot be attributed to minimizing the norm of the weights alone.

While the contribution above is interesting, it is unfortunate that the paper digresses to unrelated topics and is quite hard to understand in a few places. Some of its claims can be misleading as well. For example, the title of the paper and its abstract suggest that regularization during training is "required" but the authors only show that there exists certain forms of regularization that are *sufficient* but not necessary. In fact, as the authors mention, a simple technique like temperature scaling performs equally well as their proposed method. In addition, some of the claimed contributions are not novel or new. It is known that probability scores of neural networks are not well-calibrated because the cross entropy loss and the expressive power of neural nets lead neural nets to make overconfident predictions on the training sample. So, the first contribution stated in Page 2 is not a new contribution to be precise.

In Section 2, the authors look into the fact neural nets make confident predictions on the training sample. This entire section is chatty, imprecise, and is not necessary for the main contribution of the paper (which is in Section 3). Some issues with Section 2 include:
- The authors speak of "optimal confidence" without defining it. I originally thought that the optimal confidence of predicting y=1 is itself the true probability p(y=1|x) but Figure 2(c) shows that the authors have a different definition in mind, which they do not explain.
- The authors use a cyclic argument to explain why neural nets are overly confident on their predictions. In the paragraph below Eq 3, he authors argue that given that *we know* that neural nets perfectly fits the training examples, the upper bound on the loss in Eq 3 suggests that the predictions must match with the true labels. But that's a cyclic argument because you used the conclusion as a premise.
- There is a lengthy discussion about a "divisor" (alpha) that is not defined formally in the paper.
- In the text, the authors state that the x-axis of the Figure 2(c) is the test accuracy, written as E_{y|x} 1_y(m_x), but the legend says it is the "correct probability." Which one is it?
- What is LL in Figure 2(a)? I could not find it in the text or the figure caption.

As mentioned earlier, the primary contribution is Section 3.2. The experimental results are interesting. But, the authors later mention that a simple technique like temperature scaling would achieve similar results. Given that temperature scaling does not require any hyper-parameter to tune, I am not sure if the proposed method would be useful in practice. Unfortunately, the authors do not show how sensitive their results are to their hyper-parameters.

Regarding out-of-distribution detection, I personally do not agree that it should be used as a method for evaluating confidence scores. Suppose the data comes from a mixture of two Gaussians and you learn the mixture using expectation maximization. That algorithm would give good estimates of the probability scores because it estimates the means and covariance matrices of two Gaussian densities. However, if the distribution changes, i.e. we now have a different mixture of Gaussians, then the algorithm would still make many confident predictions that are wrong simply because the distribution has changed. That does not mean the algorithm is not well-calibrated to the distribution it was trained on.

In summary, the paper makes interesting contributions. It proposes a new method of learning well-calibrated probability scores using new forms of regularization but it is suitable for publication in its present form.

---

> ### Author Response · Authors · 2020-11-25
> **Response to reviewer 1**
>
> We appreciate reviewer 1 for the carefully reading our work and providing detailed and thoughtful feedback. We believe the the comments of reviewer 1 significantly improve our manuscript, which are addressed as follows:
>
> \# It is known that probability scores of neural networks are not well-calibrated because the cross entropy loss and the expressive power of neural nets lead neural nets to make overconfident predictions on the training sample. So, the first contribution stated in Page 2 is not a new contribution to be precise.
>
> Thanks for clarifying this. We have removed this part in the revised manuscript.
>
> \# There is a lengthy discussion about a "divisor" (alpha) that is not defined formally in the paper.
>
> We sincerely appreciate your careful review of the manuscript. We have completely removed this part in the revised manuscript, which we believe significantly more clear than the original paper.
>
> \# The authors speak of "optimal confidence" without defining it. I originally thought that the optimal confidence of predicting y=1 is itself the true probability p(y=1|x) but Figure 2(c) shows that the authors have a different definition in mind, which they do not explain.
>
> We are sorry for confusing reviewer 1. The optimal confidence in the context of Figure 2(c) was the confidence that maximizes the log-likelihood given fixed accuracy and fixed \alpha_{x,y}. However, this figure has been removed as the lengthy and unclear discussion about a divisor is removed.
>
> \# The authors use a cyclic argument to explain why neural nets are overly confident in their predictions. In the paragraph below Eq 3, the authors argue that given that we know that neural nets perfectly fit the training examples, the upper bound on the loss in Eq 3 suggests that the predictions must match with the true labels.
>
> Sorry for confusing reviewer 1. The original manuscript failed to clarify the log-likelihood on training samples and validation (or test) samples. In the revised manuscript, we carefully distinguish the log-likelihood on training samples from the behavior on the validation samples by explaining the implication of Eq. 3 as follows:
>
>
> 1) show the log-likelihood is the lower bound of the probabilistic measure of calibration.
> 2) explain that convergence of log-likelihood towards zero, for which the deep neural network can achieve, makes the predictive behavior of correcting all training samples with perfect confidence.
> 3) Observe the predictive confidence on unseen samples are significantly similar to the predictive confidence on training samples compared to the accuracy, which result in the mis-calibrated prediction.
> 4) the observation made in 3) motivates that decreasing the predictive confidence on training samples will also reduce the confidence on unseen samples, thereby reducing the calibration error.
>
>
> \# What is LL in Figure 2(a)? I could not find it in the text or the figure caption.
>
> Thanks for pointing out the missing detail. That is log-likelihood! We have added a description in the caption.
>
> \# The authors later mention that a simple technique like temperature scaling would achieve similar results. Given that temperature scaling does not require any hyper-parameter to tune, I am not sure if the proposed method would be useful in practice.
>
> As per reviewer 3, one appealing aspect of temperature scaling is the absence of hyperparameter, unlike the regularization-based approach. However, it has some drawbacks compared to the regularization-based approach. First, it requires an additional hold-out dataset. Therefore, it may not be applicable in more general settings such as continual learning and online learning. In addition, In our experiments, regularization-based approaches provided a small but consistent gain in generalization performances, while the temperature scaling does not impact the generalization performances. To clearly explain this, we have added this discussion in the conclusion.
>
>
> \# Regarding out-of-distribution detection, I personally do not agree that it should be used as a method for evaluating confidence scores.
>
> Thanks for this insightful comment. The purpose of using out-of-distribution samples was neither OOD generalization nor robustness of the model on OOD samples. Instead, we used OOD samples to investigate how the model behaves on the samples that it does not know.
> Inspired by this insightful comment, we have changed the experiment into separately measuring the predictive uncertainty on 1) correctly classified in-distribution samples, 2) misclassified in-distribution samples, and 3) OOD samples. We believe that this modified experimental setting will more clearly represent the purpose of the predictive uncertainty comparison.

---

### Official Review · AnonReviewer4 · 2020-10-27
**Review of Paper 1877**

**Rating:** 4
**Confidence:** 4

**Review:**

There has been an ongoing debate about the role and importance of explicit and implicit regularization in deep learning. This paper attempts to address this issue by arguing that explicit regularization is required for the generalization of predictive probabilities, which may not be observed under the 0-1 loss. The paper provides some discussion and numerical evidence to support the claim.

Although the paper makes some interesting remarks, the idea of looking at the predictive probability rather than classification error is certainly not new. This may be part of the story, but is unlikely to a central one. In particular, this does not explain why overparametrized models may still generalize well in the regression setting. The only theoretical analysis in the paper is based on eq. (3), which is almost trivial and does not reflect any characteristic of deep models. I find the arguments totally heuristic and not convincing enough to justify the necessity of explicit regularization. In terms of methodology, the paper examines only complexity control through weight decay and two existing explicit regularizers and does not propose any new regularization strategy.

Minor comments:
1) The predictive probability should not be referred to as “predictive confidence.” In particular, “level of predictive confidence” and “confidence control” may be confused with the standard use of “confidence” in “level of confidence” and “confidence interval.”

2) The coined terms “stochastic switch” and “deterministic score” are also misleading. The score usually means the derivative of the log-likelihood.

Update: I appreciate the authors’ response and, in particular, providing more explanations about eq. (3) (now eq. (5)). However, the explanations are still largely heuristic and based on unproven claims. To reflect the authors’ efforts to improve the paper, I am increasing my rating from 3 to 4.

---

> ### Author Response · Authors · 2020-11-25
> **Response to reviewer 4**
>
> We appreciate reviewer 4 for giving constructive and insightful feedback. We address your concerns as follows:
>
> \# Although the paper makes some interesting remarks, the idea of looking at the predictive probability rather than classification error is certainly not new.
>
> As reviewer 4 commented, the idea of looking at the predictive probability is nothing new, and recently there have been several solutions to deal with this problem such as temperature scaling and an ensemble method. However, we believe this paper’s finding could be of interest to the community as the regularization-based approach can be applied to many domains in a plug-and-play fashion and is significantly efficient compared to marginalization-based approaches such as MC dropout and deep ensemble.
>
>
>
> \# The only theoretical analysis in the paper is based on eq. (3), which is almost trivial and does not reflect any characteristic of deep models. I find the arguments totally heuristic and not convincing enough to justify the necessity of explicit regularization.
>
> Thanks for this insightful comment. As reviewer 4 points out, the eq. (3) itself does not reflect any characteristic of deep models. Besides, in the original manuscript, we did not clearly explain the implication of decomposition in eq. (3) for the deep neural network. Therefore, in the revised manuscript, we have very carefully explained this implication; as the high-capacity deep neural network can approach the maximum of the log-likelihood, it makes the predictive behavior of correcting all samples with perfect confidence. We also clearly explain how this behavior can result in overconfident predictions on unseen samples from our empirical observation.
>
>
> \# In terms of methodology, the paper examines only complexity control through weight decay and two existing explicit regularizers and does not propose any new regularization strategy.
>
>
> As per reviewer 4's comment, we opted to employ a general form of explicit regularization in the literature since our intention was to show the efficacy of explicit regularization on improving the quality of the predictive probability. For this purpose, we only slightly modify the existing method (projected error function regularization) that was proposed for regularizing the representation. However, in this paper, we apply them to constrain the maximum entry of predictive probability by applying it to only the final layer. We also explore a method to direct computation of 1D-Wasserstein distance, unlike the previous work.
>
> \# The predictive probability should not be referred to as “predictive confidence.” In particular, “level of predictive confidence” and “confidence control” may be confused with the standard use of “confidence” in “level of confidence” and “confidence interval.”
>
> We sincerely thank reviewer 4 providing this insightful comment. We agree with the reviewer’s concern that predictive confidence may be confused with the use of “confidence” in statistics, so we have added a footnote for warning the confusion.
>
> \# The coined terms “stochastic switch” and “deterministic score” are also misleading. The score usually means the derivative of the log-likelihood.
>
> Thanks for your insightful comment. We have removed the terms as these terms can make confusion.

---

### Official Review · AnonReviewer3 · 2020-10-28
**The paper study the affects of explicit regularization on calibration, although the analysis and experiments should be improved**

**Rating:** 5
**Confidence:** 3

**Review:**

The paper studies how explicit regularization affects the reliable predictive probability (calibration) of classification tasks. An analysis of the log-likelihood is presented which motivates the use of explicit regularization. Next, two regularization terms are proposed to improve the predictive probability, and experiments on CIFAR – 10/100 show that these regularization terms improve both the accuracy and the calibration error.
I think that studying how different regularization techniques affects calibration is an interesting direction. The empirical results of this paper look promising and show that indeed adding explicit regularization terms improves both the accuracy and the calibration.
I do have some concerns regarding the theoretical analysis, and the experimental setting:
1) Regarding section 2 (theoretical analysis), I am not sure how this analysis motivates that the cause of unreliable predictive probability is due to the lack of regularization. In more details:
a. The analysis of the log-likelihood is very unclear. How is the inequality of eq (3) turned into an equality? What is the \alpha_{x,y} term, is there a formula that defines it?
b. It is claimed that improving the expectation in the third sentence of page 4 is challenging due to many complex factors, therefore it is not directly connected to improving this expectation on unseen samples. Why is it true?
c. The temperature scaling method introduced there is used to improve calibration, how it is related to explicit regularization?
d. What is the bottom line of this section? In other words, how the analysis presented there motivates that what should help the calibration is explicit regularization (which seems to be the main claim of the paper)?
2) I find the experimental setup a bit lacking in two aspects:
a. The experiments are done only on CIFAR – 10 and CIFAR – 100. Since this is mainly an empirical paper, I would expect that the claims would be tested on more datasets, especially the experiments presented in Table 1.
b. The only method that is compared to the proposed regularization terms is Vanilla training. It is hard to evaluate this way whether the proposed regularization terms are really what best improves the calibration. I think it would be better to compare in Table 1 to other methods such as L_1 regularization, dropout, temperature scaling, or other known regularization/calibration techniques.

To conclude, I suggest widening the experimental setting of the paper by testing on more datasets and comparing it to other known techniques, otherwise, it is not clear whether the techniques proposed in this paper really improve on other known techniques. Regarding the theoretical analysis, I suggest making it clearer by adding rigorous theorems/propositions and explicitly stating how the analysis suggests that explicit regularization can improve the calibration.

---

> ### Author Response · Authors · 2020-11-25
> **Response to reviewer 3**
>
> We thank Reviewer 2 for your time and efforts to suggest thoughtful suggestions that significantly improve our work. We have addressed your valuable suggestions and comments as follows:
>
> \# The analysis of the log-likelihood is very unclear. How is the inequality of eq (3) turned into an equality? What is the \alpha_{x,y} term, is there a formula that defines it?
>
> We really appreciate your careful review. In the original manuscript, we presume the existence of \alpha_{x,y} since max_{k != m} log phi_k (x) <= log (1 - phi_mx (x)). However, we have removed this part as it is detrimental to clarity.
>
>
> \# It is claimed that improving the expectation in the third sentence of page 4 is challenging due to many complex factors, therefore it is not directly connected to improving this expectation on unseen samples. Why is it true?
>
> Thanks for pointing out this. The expectation indicates the generalization accuracy, and we tried to give a detailed explanation of why the generalization gap occurs and is irreducible. However, we have removed this part for clarity.
>
>
> \# The temperature scaling method introduced there is used to improve calibration, how is it related to explicit regularization?
>
> The temperature scaling maximizes the log-likelihood on unseen samples. As shown on page 3 in the new manuscript, log-likelihood on unseen samples increases the lower bound of the divergence between accuracy and a maximum entry of predictive probability; in the case of the deep neural network, this process reduces the predictive confidence to match it to its accuracy. On the other hand, the explicit regularization constrains the predictive confidence on training samples so that it produces lower confidence on unseen samples as well, which would be closer to its accuracy on unseen samples. We have added this explanation to section 2.2.
>
>
> \# How the analysis presented there motivates that what should help the calibration is explicit regularization?
>
> By eq.3 in the paper, we wanted to show that i) cross-entropy minimization naturally induces the overconfident behavior on unseen samples and ii) reducing confidence on training samples during training can help to reduce the gap between the model’s accuracy and the model’s confidence on unseen samples, i.e., the calibration error. However, we admit that we failed to present this point clearly in the original manuscript. Therefore, we have thoroughly revised section 2.2 (the motivation section) in the following way:
>
>
> 1) show the log-likelihood is the lower bound of the probabilistic measure of calibration.
> 2) explain that convergence of log-likelihood towards zero, for which the deep neural network can achieve, makes the predictive behavior of correcting all training samples with perfect confidence.
> 3) Observe the predictive confidence on unseen samples are significantly similar to the predictive confidence on training samples compared to the accuracy, which result in the mis-calibrated prediction.
> 4) the observation made in 3) motivates that decreasing the predictive confidence on training samples will also reduce the confidence on unseen samples, thereby reducing the calibration error.
>
>
> We believe that the revised manuscript now clearly explains the impact of the cross-entropy minimization on overconfident behavior and the necessity of reducing confidence.
>
>
> \# The experiments are done only on CIFAR – 10 and CIFAR – 100. Since this is mainly an empirical paper, I would expect that the claims would be tested on more datasets.
>
> Thanks for the great suggestion. In response to reviewer 3, we performed additional experiments on a document classification task with pre-trained BERT. As consistent with results on the image classification task, all explicit regularization methods greatly improve the calibration performances as well as provide a consistent gain in test accuracy. We believe that this additional result adds value to our findings.
>
>
>
> \# The only method that is compared to the proposed regularization terms is Vanilla training. It is hard to evaluate this way whether the proposed regularization terms are really what best improves the calibration. I think it would be better to compare in Table 1 to other methods such as L_1 regularization, dropout, temperature scaling, or other known regularization/calibration techniques.
>
> Thanks for this insightful comment. In the original manuscript, we focused only on showing the effects of explicit regularization to highlight the role and importance of regularization in deep learning. Therefore, we omitted the extensive comparison with existing methods (except temperature scaling on CIFAR-100 provided for baseline performance) as our goal was not to achieve SOTA in improving calibration. However, we have added a new experiment on misclassification/OOD detection tasks during the discussion period, in which the regularization-based approach is compared to the deep ensemble and the MC dropout.

---

### Official Review · AnonReviewer2 · 2020-10-28
**many technical inaccuracies**

**Rating:** 3
**Confidence:** 3

**Review:**

This paper aims to improve the "reliable predictive probability" when using a single deterministic deep net. The paper proposes explicit regularization methods to achieve this goal.

Overall, I found the paper hard to follow, often unclear in what the goal is, jumping quickly from one point to another one. Also many crucial technical steps seem incorrect or at least not well motivated/discussed. A few examples :

1. What do the authors exactly mean by “reliable” predictive probabilities, which is at the core of this paper? In the introduction, calibration, overconfidence of the model and uncertainty representation (like in Bayesian statistics) are discussed. Does “reliability” refer to all of this at once, or is it some other concept? How is it defined? Given that this is not clear to me, it makes it hard for me to understand if  the paper provides some approaches to improve it.

2. The inequality in Eq 3 could be discussed in some detail. My understanding is that each \phi_k is replaced by 1-\phi_mx in the second part, i.e., the maximum possible probability 1-\phi_mx is used for each misclassification. This seems very extreme to me.  Moreover, these “new probabilities“ do not sum to one any longer: \phi_mx + (K-1) * (1-\phi_mx) = 1+ (K-2) * (1-\phi_mx) >> 1, which grows linearly with the number of classes K (if \phi_mx <1), which seems weird. I wonder if this extremely loose bound itself causes the high cost of misclassification that is later “derived” as a key problem.

3. It would also be good to discuss why it is useful to consider an upper bound on a quantity that gets maximized (log likelihood). Maximizing a lower bound would seem more natural.

4. The divisor \alpha_x,y is introduced in Eq 3 as a mechanism to achieve equality. As I understand Eq 3, however, \alpha_x,y cannot be a function of y at the location where it is introduced in Eq 3, as it is  outside of the expectation E_y|x. It would be good to clarify this before deriving further insights.

5. The L2-norm of function f^W is discussed in several places before it is defined in Eq 5.

6. Below Eq 4, \tau^\star is defined in a maximization problem. However,  \tau can be moved outside of the expectation: E[ \log(\phi/\tau)]=  E[\log(\phi)] - \log(\tau), hence this is trivially maximized when \tau approaches 0, which does not make sense to me.

Apart from that, also the language of the paper could be greatly improved.

---

> ### Author Response · Authors · 2020-11-25
> **Response to reviewer 2**
>
> We sincerely appreciate reviewer 2 for the carefully reading our work and valuable comments. We address your concerns as follows:
>
> \# What do the authors exactly mean by “reliable” predictive probabilities, which is at the core of this paper?
> Thanks for pointing out an important missing detail. By reliable predictive probability, we mean both well-calibrated prediction and a better predictive uncertainty representation. To clarify this, we have added the following paragraph on page 1: “In this paper, we call predictive probability reliable if it is well-calibrated and precisely represents uncertainty about its predictions.”
>
>
> \# The inequality in Eq 3 could be discussed in some detail. My understanding is that each \phi_k is replaced by 1-\phi_mx in the second part, i.e., the maximum possible probability 1-\phi_mx is used for each misclassification. This seems very extreme to me. Moreover, these “new probabilities“ do not sum to one any longer: \phi_mx + (K-1) * (1-\phi_mx) = 1+ (K-2) * (1-\phi_mx) >> 1, which grows linearly with the number of classes K (if \phi_mx <1), which seems weird. I wonder if this extremely loose bound itself causes the high cost of misclassification that is later “derived” as a key problem.
>
> We are sincerely sorry for confusing reviewer 2. In the original manuscript, the aggregated term (\sum_{k != m_x} 1_{y}(k) phi_k) was replaced by 1-\phi_mx (which holds because we consider only a single label detection problem). In the revised manuscript, we explicitly state how inequality can be derived.
>
>
> \# It would also be good to discuss why it is useful to consider an upper bound on a quantity that gets maximized (log likelihood). Maximizing a lower bound would seem more natural.
>
> In the original manuscript, we tried to illustrate which factors cause the reduced log-likelihood on unseen samples that get maximized during training. However, as this is not clearly explained in the original manuscript and prune to make confusion, we thoroughly revised the motivation part as follows:
>
> 1) show the log-likelihood is the lower bound of the probabilistic measure of calibration.
> 2) explain that convergence of log-likelihood towards zero, for which the deep neural network can achieve, makes the predictive behavior of correcting all training samples with perfect confidence.
> 3) Observe the predictive confidence on unseen samples are significantly similar to the predictive confidence on training samples compared to the accuracy, which result in the mis-calibrated prediction.
> 4) the observation made in 3) motivates that decreasing the predictive confidence on training samples will also reduce the confidence on unseen samples, thereby reducing the calibration error.
>
>
> \# The divisor \alpha_x,y is introduced in Eq 3 as a mechanism to achieve equality. As I understand Eq 3, however, \alpha_x,y cannot be a function of y at the location where it is introduced in Eq 3, as it is outside of the expectation E_y|x. It would be good to clarify this before deriving further insights.
>
> Thanks for your careful review. In the original manuscript, we presume the existence of \alpha_{x,y} since max_{k != m} log phi_k (x) <= log (1 - phi_mx (x)). However, we have removed this part as it is detrimental to the clarity.
>
> \# The L2-norm of function f^W is discussed in several places before it is defined in Eq 5.
>
> Thanks for pointing out this. We have added a brief explanation in a footnote with a comment that the L^p norm will be discussed in detail in section 3.2.
>
>
> \# Below Eq 4, \tau^\star is defined in a maximization problem. However, \tau can be moved outside of the expectation: E[ \log(\phi/\tau)]= E[\log(\phi)] - \log(\tau), hence this is trivially maximized when \tau approaches 0, which does not make sense to me.
>
> We sincerely thank you for your careful review. E[ log(\phi/\tau)] was a typo, and it has been corrected to E[ log(\sigma(f(x)/tau))], i.e., dividing logits by a tau before applying softmax.
>
> \# The language of the paper could be greatly improved.
>
> Thanks for your careful review. In response to reviewer X, we have significantly improved the clarity and corrected grammar errors by carefully proofreading the manuscript.

---

### Official Review · AnonReviewer5 · 2020-11-04
**Interesting result, well-grounded theoretically; needs empirical results on more datasets/modalities**

**Rating:** 5
**Confidence:** 4

**Review:**

Update: After reading the other reviews/responses, I think there are persistent concerns with the breadth of experiments and the substantiveness of the contribution; although the manuscript is somewhat improved by the authors' updates, I'm keeping my score at 5.

This paper examines the effect of explicit regularization on the quality of predicted probabilities from neural network classifiers. Empirical results show that explicit regularization substantially improves predictive uncertainty on CIFAR10/100, as measured by Negative Log Likelihood and Expected Calibration Error. Additional results demonstrate the effectiveness of explicit regularization on separating out-of-distribution (OOD) data, with CIFAR as in-distribution and SVHN as OOD.

The paper is well-grounded theoretically, and well-written and organized. The experiments show convincing results on the ability of explicit regularization to improve the quality of uncertainty of neural network classifiers. The baselines chosen are sensible (temperature scaling for an in-distribution test set, ensembles/MC dropout/temperature scaling for the OOD tasks).

The datasets and modalities tested are very limited, and the paper would be stronger if it showed experimental results on additional datasets besides CIFAR and SVHN, and additional modalities besides images. In particular, Nalisnick et al. 2018 (“Do deep generative models know what they don’t know?”) showed that models trained on CIFAR assign high likelihood to examples from SVHN, which I think calls into question how well results on this pair of datasets generalize.

The entropy diagrams in Figure 4 give a good qualitative indication of OOD detection, but the results would be strengthened by a quantitative measure such as AUROC/AUPRC. Also, Figure 4 is missing a legend.

I believe the title overreaches a little -- the results of the paper are interesting and compelling, but they fall short of demonstrating that regularization in deep learning models is “required” for reliable predictive probability.

---

> ### Author Response · Authors · 2020-11-25
> **Response to reviewer 5**
>
> We appreciate reviewer 5 for providing valuable suggestions. We address your comments as follows:
>
> \# The datasets and modalities tested are very limited.
>
> In response to reviewer 5, we have added new experiments on a document classification task with pre-trained BERT. In the new experiments, we observed that all explicit regularization methods produce results consistent with findings in the previous image classification tasks.
>
>
> \# The entropy diagrams in Figure 4 give a good qualitative indication of OOD detection, but the results would be strengthened by a quantitative measure such as AUROC/AUPRC.
>
> Thanks for the great suggestions. We have added quantitative comparison in the revised manuscript, which shows that all regularization methods greatly improved OOD/misclassification detection task performances.
>
> \#  Figure 4 is missing a legend.
>
> Thanks. We have added a legend.
>
> \# I believe the title overreaches a little -- the results of the paper are interesting and compelling, but they fall short of demonstrating that regularization in deep learning models is “required” for reliable predictive probability.
>
> Thanks for pointing out this. We agree with the reviewer's comment that the original title is somewhat exaggerated. Therefore, we have changed the title to “Revisiting explicit regularization in neural networks for reliable predictive probability,” which we believe is more appropriate.

---

### Author Response · Authors · 2020-11-25
**Summary of responses**

We sincerely thank the reviewers for their insightful comments on improving our manuscript. We believe that these thoughtful and constructive criticisms make our manuscript significantly better. We have uploaded the revised manuscript addressing the feedback given by reviewers, which contains the following main changes:

1. We thoroughly modified the section 2 of the manuscript in order to clearly explain the motivation of explicit regularization for producing reliable predictive probability.
2. We performed additional extensive experiments on document classification with BERT.
3. We added quantitative evaluation of explicit regularization methods on misclassification/OOD detection tasks.

---

### Decision · Program_Chairs · 2021-01-07
**Final Decision**

**Decision:**

Reject

**Comment:**

Summary of reviews and discussions: Reviewers were overwhelmingly negative on this paper due to a variety of factors: unclear writing, heuristic motivation, overpromising in the title while underdelivering on results. Although the authors responded to the reviewers' feedback, and some reviewers increased their score to acknowledge the author's work, they remained unconvinced overall and no reviewer argued strongly for acceptance.